# LRH-1 agonism favours an immune-islet dialogue which protects against diabetes mellitus

Nadia Cobo-Vuilleumier[1], Petra I. Lorenzo[1], Noelia García Rodríguez[1], Irene de Gracia Herrera Gómez[1], Esther Fuente-Martin[1], Livia López-Noriega[1], José Manuel Mellado-Gil[1], Silvana-Yanina Romero-Zerbo[2,3], Mathurin Baqué[4], Christian Claude Lachaud[1], Katja Stifter[5], German Perdomo[6], Marco Bugliani[7], Vincenzo De Tata[8], Domenico Bosco[9], Geraldine Parnaud[9], David Pozo [10], Abdelkrim Hmadcha [1,3], Javier P. Florido[11], Miguel G. Toscano[12], Peter de Haan[12], Kristina Schoonjans[13], Luis Sánchez Palazón[14], Piero Marchetti[7], Reinhold Schirmbeck[5], Alejandro Martín-Montalvo[1], Paolo Meda[15], Bernat Soria[1,3], Francisco-Javier Bermúdez-Silva [2,3], Luc St-Onge[16] & Benoit R. Gauthier [1]

Type 1 diabetes mellitus (T1DM) is due to the selective destruction of islet beta cells by immune cells. Current therapies focused on repressing the immune attack or stimulating beta cell regeneration still have limited clinical efficacy. Therefore, it is timely to identify innovative targets to dampen the immune process, while promoting beta cell survival and function. Liver receptor homologue-1 (LRH-1) is a nuclear receptor that represses inflammation in digestive organs, and protects pancreatic islets against apoptosis. Here, we show that BL001, a small LRH-1 agonist, impedes hyperglycemia progression and the immune-dependent inflammation of pancreas in murine models of T1DM, and beta cell apoptosis in islets of type 2 diabetic patients, while increasing beta cell mass and insulin secretion. Thus, we suggest that LRH-1 agonism favors a dialogue between immune and islet cells, which could be druggable to protect against diabetes mellitus.

[1] Department of Cell Regeneration and Advanced Therapies, Andalusian Center for Molecular Biology and Regenerative Medicine-CABIMER, Junta de Andalucia-University of Pablo de Olavide-University of Seville-CSIC, Seville 41092, Spain. [2] Unidad de Gestión Clínica Intercentros de Endocrinología y Nutrición, Instituto de Investigación Biomédica de Málaga (IBIMA) Hospital Regional Universitario de Málaga Universidad de Málaga, Málaga 29010, Spain. [3] Centro de Investigación Biomédica en Red de Diabetes y Enfermedades Metabólicas Asociadas (CIBERDEM), Madrid 28029, Spain. [4] Neurix SA, Geneva 1228, Switzerland. [5] Ulm University Hospital, Ulm 89081, Germany. [6] Facultad de Ciencias de la Salud, Universidad de Burgos, Burgos 09001, Spain. [7] Department Clinical and Experimental Medicine, University of Pisa—AOUP University Hospital, Pisa 56126, Italy. [8] Department of Translational Research and of New Surgical and Medical Technologies, University of Pisa, Pisa 56126, Italy. [9] Cell Isolation and Transplantation Centre, University Hospital, Geneva 1211, Switzerland. [10] Department of Cell Dynamics and Signalling, CABIMER-Andalusian Center for Molecular Biology and Regenerative Medicine, Seville 41092, Spain. [11] Clinical Bioinformatics Area, Fundación Progreso y Salud, Consejería de Salud, Seville 41013, Spain. [12] Amarna Therapeutics, Seville 41092, Spain. [13] Laboratory of Metabolic Signaling, EPFL, Lausanne 1015, Switzerland. [14] Biological Resources, CABIMER-Andalusian Center for Molecular Biology and Regenerative Medicine, Seville 41092, Spain. [15] Department of Cell Physiology and Metabolism, University of Geneva, Geneva 1211, Switzerland. [16] Neuried Munich 82061, Germany. Correspondence and requests for materials should be addressed to B.R.G. (email: benoit.gauthier@cabimer.es)

Type 1 diabetes mellitus (T1DM) is a CD4[+] and CD8[+] T-cell-dependent autoimmune disease that targets beta cell destruction, ultimately leading to hyperglycemia and insulin dependence. The collapse in tolerance to self-antigens, such as insulin, is precipitated by genetic and environmental factors[1,2]. To date, therapies aimed at inhibiting the immune system using anti-CD3 monoclonal antibodies or at neutralizing pro-inflammatory cytokines, have had limited success[3,4]. One of the reasons may be that inhibiting the immune and inflammatory reactions in the pancreas impairs the repairing and regeneration capabilities of a functional beta cells mass[5,6], as observed during wound healing[7]. Novel agents that could guide a pro-inflammatory autoimmune destructive environment toward an anti-inflammatory milieu facilitating islet regeneration, would define a novel class of antidiabetic therapies.

The liver receptor homolog-1 (LRH-1, or NR5A2) is a member of the NR5A family of nuclear receptors, which plays a pivotal role in early embryonic development, and specifies the endodermal lineage[8]. In the liver, LRH-1 modulates the expression of genes involved in cholesterol and bile acid metabolism, as well as in glucose homeostasis[9], attenuates the hepatic acute phase response, which is triggered upon increases of pro-inflammatory cytokines, and protects against endoplasmic reticulum stress[10,11]. In the intestine, LRH-1, modulates the enterocyte renewal and regulates the local immune system via production of glucocorticoids[12]. In the pancreas, LRH-1 regulates the expression of genes involved in digestive functions, and protects the endocrine islets against cytokine- and streptozotocin-induced apoptosis[13,14], while stimulating the production of enzymes involved in glucocorticoids biosynthesis[14]. In view of the above, specifically of the possibility that LRH-1 could elicit an islet-driven anti-inflammatory microenvironment, we posited that upregulating LRH-1 activity could have beneficial therapeutic effects in diabetes mellitus (DM).

Natural phospholipids physiologically stimulates LRH-1 activity[15,16], decreasing hepatic steatosis and improving glucose homeostasis in animal models of insulin resistance[17]. Given that LRH-1 can also be activated by smaller, non-polar bicyclic compounds[18], we have synthesized a compound termed BL001, which we have tested in mouse models of T1DM, as well as in pancreatic islets from patients affected by Type 2 DM (T2DM). Here we report that the long-term in vivo administration of BL001 prevents the development of diabetes in mice, through the combined maintenance of a functional islet beta cell mass and the release of anti-inflammatory factors, which contribute to the islet regeneration effect. We further report that BL001 also protects human islet cells from apoptosis and improves impaired insulin secretion as well as beta cell survival in the pancreatic islets of T2DM patients. The data define LRH-1 as a novel therapeutic target for the treatment of T1DM.

## Results

**BL001 activates LHR-1 without cytotoxic or metabolic effects.** The chemical structure of BL001, which specifically binds to and activates LRH-1[18], is depicted in Supplementary Fig. 1a. The effects of the drug on LRH-1 activity, cell viability, and toxicity are described in Supplementary Fig. 1b–e. Pharmacokinetic and safety profiling of BL001 were studied in C57BL/6 and RIP-B7.1 mice, respectively. An i.p. injection of 10 mg/kg b.w. BL001 led to peak plasma concentrations of 3.6 μg/ml (≈8 μM) after 0.2 h, and a half-life of 9.4 h. Daily injections during 24 weeks did not reveal macroscopic organ alterations in BL001-treated RIP-B7.1 mice (Supplementary Fig. 2a, b), which also featured normal plasma levels of total cholesterol and triglycerides up to 8 weeks of treatment (Supplementary Fig. 3a, b). Insulin sensitivity was not altered by this BL001 treatment (Supplementary Fig. 3c).

**BL001 blunts apoptosis and attenuates diabetes in mice.** To assess the anti-apoptotic effect of BL001, mouse islets were exposed to 10 μM BL001 in the presence of 2 ng/ml IL1beta, 28 ng/ml TNFalpha and 833 ng/ml IFNgamma. The drug prevented the cytokine-induced islet cell death (Fig. 1a). A substantial loss of LRH-1 transcript and protein by RNAi, sensitized BL001-treated islets to the cytokine-induced apoptosis (Fig. 1b–d). The anti-diabetic role of BL001 was next evaluated in animal models of T1DM. C57BL/6 male mice that received 150 mg/kg b.w. streptozotocin (STZ) developed diabetes within 4 weeks (Fig. 1e, Supplementary Fig. 4a). The incidence of diabetes was decreased after a 5 day pre-treatment with 10 mg/kg b.w. BL001 (Fig. 1e, Supplementary Fig. 4a), which decreased the loss of insulin-containing beta cells (Fig. 1g), and increased the proportion of cells staining for both insulin and glucagon (Fig. 1h, i). Moreover, 30% of the mice that developed diabetes returned to normoglycemia 4 weeks after a daily injection of 10 mg/kg b.w. BL001, starting 1 week after the STZ injection (Fig. 1f, Supplementary Fig. 4b).

To evaluate the effect of BL001 against an autoimmune attack, we studied RIP-B7.1 mice, a model mimicking the etiology of T1DM[19], without gender influence[20]. Eighty percent of RIP-B7.1 mice immunized against insulin developed diabetes within 8 weeks (Fig. 1j, Supplementary Fig. 4c), when receiving only a vehicle solution. This proportion dropped to 43% in mice treated with 10 mg/kg b.w. BL001 for 5 days prior to immunization (Fig. 1j, Supplementary Fig. 4c). Still, a similar increase in the proliferation of T cells in response to insulin was also detected in the splenocytes of mice treated or not with BL001, both 4 and 8 weeks after the insulin immunization (Fig. 1k), confirming that all mice had mounted an in vivo autoimmune attack. At these time points, immunostaining showed a near-complete destruction of beta cells in the islets of the control immunized mice, and the persistence of sizable numbers of these cells in the immunized mice treated with BL001 (Fig. 1l). In the presence of hyperglycemia, the latter animals also featured islets with a significant increase in the number of PDX1[+] cells as compared to immunized mice (Fig. 1m, n).

Mice in which the BL001 treatment was initiated 5 days after the immunization, featured an incidence of diabetes similar to that of controls for the first 5 weeks (Fig. 1o, Supplementary Fig. 4d). Thereafter, however, the incidence of diabetes was lower (~30%) in the BL001-treated mice than in immunized controls (~80%; Fig. 1o, Supplementary Fig. 4d). A similar decrease in diabetes incidence was observed in female NOD mice treated with BL001 from week 12 (Fig. 1p), an age at which these animals feature an ongoing autoimmune attack, and reduced insulin content. To establish whether the in vivo effects of BL001 were mediated via LRH-1, we generated mice in which the *Lrh1* gene was selectively disrupted in beta cells. To this end, Lrh1 lox/lox mice were crossed with RIP-Cre mice. We found that all mice carrying two deleted alleles (βLRH-1[−/−]), and 75% of the progenies carrying one normal allele (βLRH-1[−/+]) died within 3 weeks of life (Supplementary Fig. 5a). Immunofluorescence staining of islets from day 1 neonatal pups, showed an atypical islet morphology, with a normal proportion of insulin-containing beta cells surrounded by an increased proportion of glucagon-containing alpha cells in homozygous mice (Supplementary Fig. 5b–e).

**BL001 inhibits insulitis and alters serum cytokine profile.** To unravel the mechanism whereby BL001 impedes the progression of diabetes, we evaluated the lymphocytic infiltration of islets (insulitis) in mice pre-treated or not with BL001. Four weeks after immunization, a sizable insulitis was detected in the pancreas of

both BL001-treated and control mice (Fig. 2a, b). However, 4 weeks later, the normoglycemic BL001-treated mice were nearly free of insulitis, in contrast to controls which displayed strong infiltration (Fig. 2c, d). Accordingly, no CD4+ or CD8+ T cells were found in the islets of the former animals (Supplementary Fig. 6), which also showed significantly higher levels of circulating

IL10, IL5, IL6 (Fig. 2e, f), CCL2, CCL4 (Fig. 2g, h), and TGFbeta (Fig. 2i, j) than immunized controls at 8 but not 4 weeks. In contrast, no significant differences were observed in levels of INFgamma ($0.6965 \pm 0.0931$ versus $0.4989 \pm 0.0733$ $p = 0.1072$, unpaired Student's $t$ test), TNFalpha ($5.925 \pm 0.815$ versus $6.648 \pm 0.659$ $p = 0.4982$, unpaired Student's $t$ test), IL1beta ($0.5267 \pm$

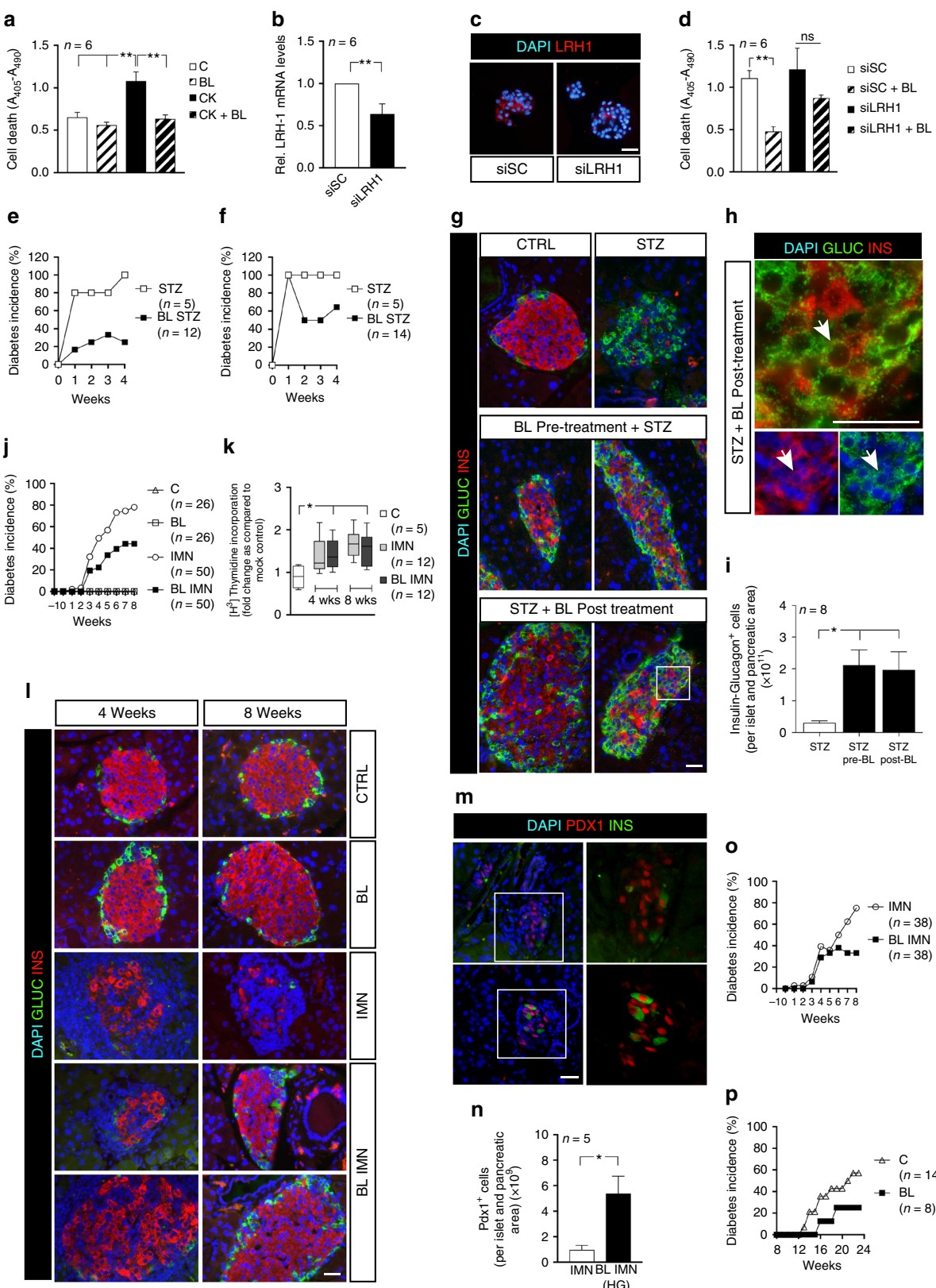

0.0913 versus 0.3461 ± 0.0579 $p = 0.1092$, unpaired Student's $t$ test) and IL2 (1.306 ± 0.181 versus 1.111 ± 0.154 $p = 0.4221$, unpaired Student's $t$ test) between immunized untreated and BL001-treated mice at 8 weeks.

**BL001 elicits a tolerogenic environment within the pancreas**. To analyze how BL001 remodeled the immune cell environment, we performed flow cytometry studies on pancreatic cell suspensions (gating strategy shown in Supplementary Fig. 7). The proportions of CD4+/CD25+/FoxP3+ Tregs and CD4+/IL4+ Th2 cells were higher in the pancreas of the immunized and BL001-treated RIP-B7.1 mice than in the cognate controls (Fig. 3a, c), whereas CD4+/IFN gamma+ Th1 cells were decreased, and CD4+/IL17+ Th17 cells were unaffected (Fig. 3b, d). Furthermore, the number of CD103+/IDO+ tolerogenic dendritic cells, F4/80+/CD11b+ macrophages, and their M2-like anti-inflammatory subtype (F4/80+/CD11b+/CD206+) were also increased in BL001-treated mice (Fig. 3e–g). Accordingly, transcriptome profiling revealed that BL001 significantly increased the expression of key genes associated with the M2 genetic signature, including, *Clec7a* (Dectin-1), *Clec10a* (CD301), *Chst7*, *Tiam1*, *Cd300ld*, and *Olfm1* (Noelin-1 or Pancortin)[21] (Fig. 3h).

**BL001 favors a M2 macrophage phenotype via LRH-1**. To further delineate the role of BL001 in promoting a M2 macrophage phenotype resulting in increased IL10 secretion, peritoneal macrophages were isolated and treated in vitro with the compound. BL001 dose-dependently increased transcript levels and the secretion of IL10 and TGFbeta (Fig. 4a–d). Exposure to LPS, also enhanced the expression and secretion of IL10, but not of TGF beta by these cells (Fig. 4e–h). To substantiate that BL001-mediated M2 polarization was conveyed by direct LRH-1 activation, the latter receptor was knock down using siRNA. Silencing of LRH-1 transcript and protein levels by ~70% in primary macrophages (Fig. 4i, j), significantly blunted the BL001-mediated increase in expression of the M2 signature genes, *Clec7a*, *Clec10a*, and *Olfm1* as well as *Il10* (Fig. 4k, l, p, q), independent of LPS treatment. Transcript levels of *Chst7*, *Tiam1*, and *Cd300ld* were also repressed, albeit in macrophages either treated or not with LPS (Fig. 4m–o). In contrast, TGFbeta expression was not altered in LRH-1 silenced cells (Fig. 4r). As IL10 biosynthesis and secretion in alternatively activated macrophage has been linked to *Clec7a* stimulation[22,23], secretion levels of this cytokine was next evaluated in LRH-1 knockdown cells. Accordingly, IL10 secretion was decreased in BL001-treated macrophages independent of LPS treatment (Fig. 4s) indicating that LRH-1 is involved in the

BL001-dependent polarization and cytokine secretory function of M2 macrophages.

**BL001 increases the number of bi-hormonal cells in islets**. To assess whether the BL001 treatment favored the survival and replenishment of islet cells, we morphometrically evaluated the mass of beta and alpha cells, as well as islet numbers and sizes. Eight weeks after immunization, all these parameters were increased in the BL001-treated mice (Fig. 5a–d, Supplementary Fig. 8a–c). This increase was not associated to enhanced proliferation or decreased apoptosis of beta cells, as assessed using different methods (Fig. 5e–h, Supplementary Fig. 9a, b). In contrast, the immunized mice, which did not receive BL001, featured higher levels of beta cell proliferation and apoptosis as a result of the ongoing autoimmune attack (Fig. 5e–h). Eight weeks after immunization, the mass of alpha cells was also increased in BL001-treated mice in the absence of enhanced proliferation (Fig. 5c, d, i, j, Supplementary Fig. 9c). At this time point, islets of these mice, also contained increased numbers of cells co-expressing insulin and glucagon, as well as PDX1 and glucagon (Fig. 5k–n and supplementary Fig. 10). Since the data suggest the potential activation of a genetic beta cell program in alpha cells, we investigated whether a 48-h BL001 treatment could suppress the expression of ARX, whose down-regulation triggers beta-to-alpha cell trans-differentiation[24]. Accordingly, a 2-day exposure to BL001 decreased the expression of ARX, glucagon, and MafB transcripts in the alpha cell-derived TC1–6 cell line (Fig. 5o).

**BL001 protects human islet from apoptosis improving function**. Having validated the proof-of-concept in mice that BL001 protects islets and prevents both chemically and autoimmune-induced diabetes, we next sought to determine whether these beneficial effects were translated to human islets under stress conditions. Ten to twenty μM BL001 did not reveal any cytotoxic effects on islets obtained from normoglycaemic donors (Supplementary Fig. 11a). In the same islets, the expression levels of LRH-1 were also unchanged, whereas that of its target gene *shp* was increased (Supplementary Fig. 11b). However, 10 μM BL001 decreased the apoptosis of islet beta cells, 24 and 72 h after an exposure to either cytokines (Fig. 6a, c) or streptozotocin (Fig. 6b, d). Glucose-stimulated insulin secretion (GSIS) of islets from non diabetic donors (Supplementary Table 1) was not modified by BL001 (Fig. 6e). In contrast, GSIS of islets from type 2 diabetic donors (Supplementary Table 2) was significantly increased after exposure to BL001 (Fig. 6f). Under these conditions, the proportion of apoptotic beta cells was significantly decreased (Fig. 6g,

**Fig. 1** BL001 improves islet survival and blunts development of diabetes in three mouse models. **a** Exposure to 10 μM BL001 (BL) for 72 h blocked cytokine-induced apoptosis (CK; 2 ng/ml IL1beta, 28 ng/ml TNFalpha and 833 ng/ml IFNgamma) in mouse islets. Cell death was assessed by ELISA quantification of mono and oligonucleosomes released by apoptotic cells. This blockade was blunted in islets transfected with siRNAs (**d**) which decreased LRH-1 transcript (**b**) and protein levels (LRH1, red; scale bar: 25 μM) (**c**). Data are means + s.e.m. values of six independent experiments using six different islet preparations and performed in triplicates. siSC, pool of siRNAs against scrambled sequence. BL001 (10 mg/kg b.w.) also decreased the incidence of diabetes in C57Bl/6 male mice treated with a single dose of streptozotocin (STZ;150 mg/kg b.w.), irrespective of whether the former drug was given prior to (**e**) or 1 week after (**f**) STZ. Values are means ± s.e.m. of $n = 5$ (STZ)-14 mice (BL/STZ). After these 2 protocols, insulin-containing beta cells (red) and glucagon-containing alpha cells (green) were preserved in islets of STZ and BL001-treated mice (**g**), which also featured an increase number of cells containing both hormones (**h**, arrowheads). Nuclei were stained with DAPI (blue). Scale bars: 25 μM. **i** Quantification of insulin- and glucagon-positive cells. **j** A 5-day BL001 pre-treatment of RIP-B7.1 mice prior to immunization decreased the diabetes incidence. Values are means ± s.e.m. C (vehicle treated, $n = 26$ male and female mice); BL (BL001, $n = 26$ male and female mice); IMN (immunized, $n = 50$ male and female mice), and BL IMN (BL001-treated and immunized, $n = 50$ male and female mice). **k** Immunization increased the proliferation of RIP-B7.1 splenocytes, an effect that was not modified by the BL001 treatment. Values are median (lines in the boxes) ± median errors of 5–12 preparations. **l** The 8-week-long BL001 treatment preserved insulin- (red) and glucagon-containing cells (green) in islets of immunized RIP-B7.1 mice (scale bar: 25 μM). **m, n** BL001-treated and immunized RIP-B7.1 mice that developed hyperglycemia contained a higher number of PDX1+ cells as compared to untreated and immunized animals (scale bar: 25 μM). **o** Diabetes incidence of RIP-B7.1 mice that were immunized at days 0 and 7 and subsequently injected daily, starting at week 1 and for up to 8 weeks, with either vehicle (IMN, $n = 38$ male and female mice) or BL001 (BL IMN, $n = 38$ male and female mice). **p** Diabetes incidence in female NOD mice receiving (BL, $n = 8$) or not (C, $n = 14$) BL001 from 12 weeks. *$p < 0.05$, **$p < 0.01$, ***$p < 0.001$, and ****$p < 0.0001$, one-way ANOVA (**a**, **d**, **k**, **i**) or Student's $t$ test (**b**, **n**)

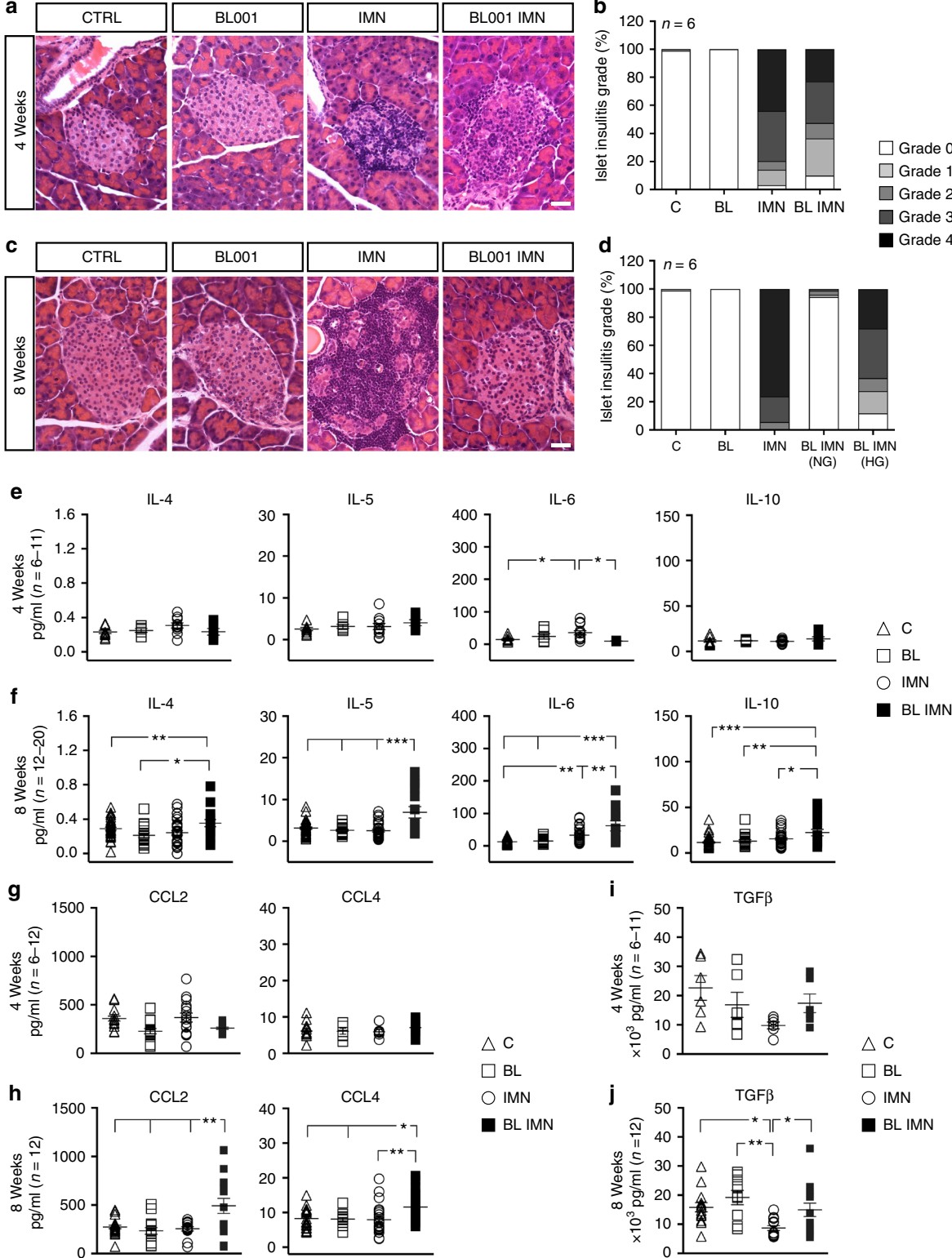

**Fig. 2** BL001 decreases insulitis and modulates the blood cytokine profile. Histology (representative images, scale bar: 25 μM) and morphometry at **a**, **b** 4 and **c**, **d** 8 weeks revealed that treatment with 10 mg/kg b.w. BL001 decreased the incidence and severity of insulitis in immunized RIP-B7.1 mice. At 8 weeks, BL001-treated and immunized mice that developed hypeglycemia exhibited a similar degree of insulitis to the BL001-treated and immunized mice that were normoglycemic at 4 weeks. Impact of BL001 treatment on the profile of cytokines, chemokines and TGFbeta at 4 weeks (**e**, **g**, **i**) and 8 weeks (**f**, **h**, **j**). Plots show individual mice values. *$p < 0.05$, **$p < 0.01$, and ***$p < 0.001$, one-way ANOVA

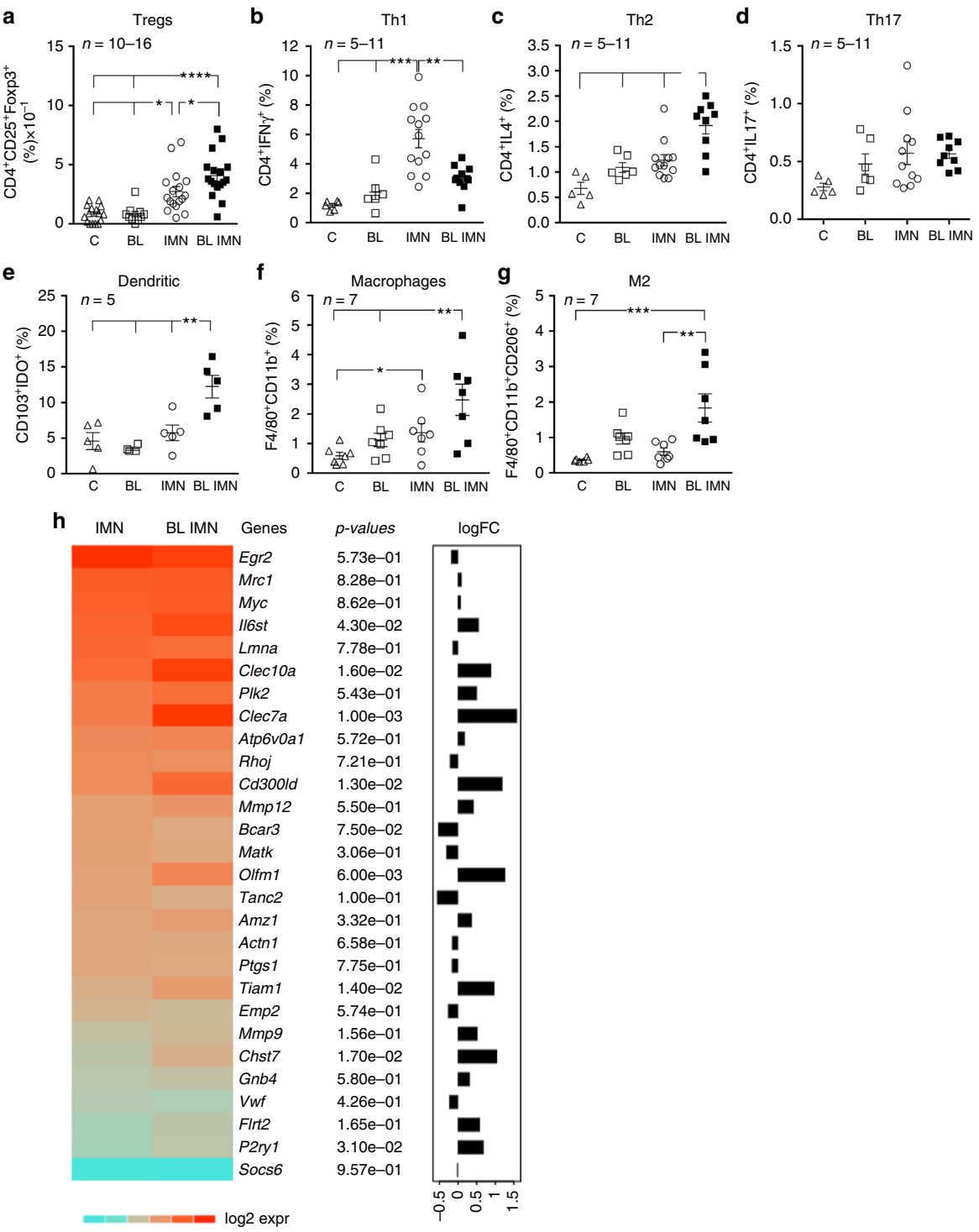

**Fig. 3** BL001 promotes a pancreatic anti-inflammatory/tolerogenic environment. RIP-B7.1 male and female mice were pre-treated or not with BL001 for 5 days and then immunized. Treatment with BL001 was pursued for an additional 8 weeks at which point animals were sacrificed and pancreas extracted and processed for flow cytometry quantification of **a** Tregs (n = 10–16 mice), **b** Th1 (n = 5–11 mice), **c** Th2 (n = 5–11 mice), **d** Th17 (n = 5–11 mice), **e** dentritic cells (n = 5 mice), **f** macrophages (n = 7 mice), and **g** M2-like macrophages (n = 7 mice). **h** Heatmap displaying expression values of a subset of M2 macrophage genes in F4/80$^+$/CD11b$^+$/CD206$^+$ pancreatic subpopulations isolated from immunized mice treated or nor with BL001 (IMN and IMN BL). Also depicted are raw p-values and the graphical representation of logFC from the differential expression analysis (IMN BL versus IMN). *p < 0.05, **p < 0.01, ***p < 0.001, and ****p < 0.0001, one-way ANOVA (**a**–**g**)

h) and paralleled by reduced cleaved caspase-3 activity (Fig. 6i, j and Supplementary Fig. 12).

To assess whether comparable effects could be observed during an ongoing, in vivo immune attack, mimicking a T1DM environment, we transplanted human islets under the kidney capsule of immune competent C57BL/6 mice, treated or not with BL001. Two days after xenotransplantation, mice were treated with daily injections of BL001 or vehicle for 7 days. At this time point, rejection of the human islets is anticipated[25,26]. Consistent with the protective effect of BL001, grafts from BL001-treated

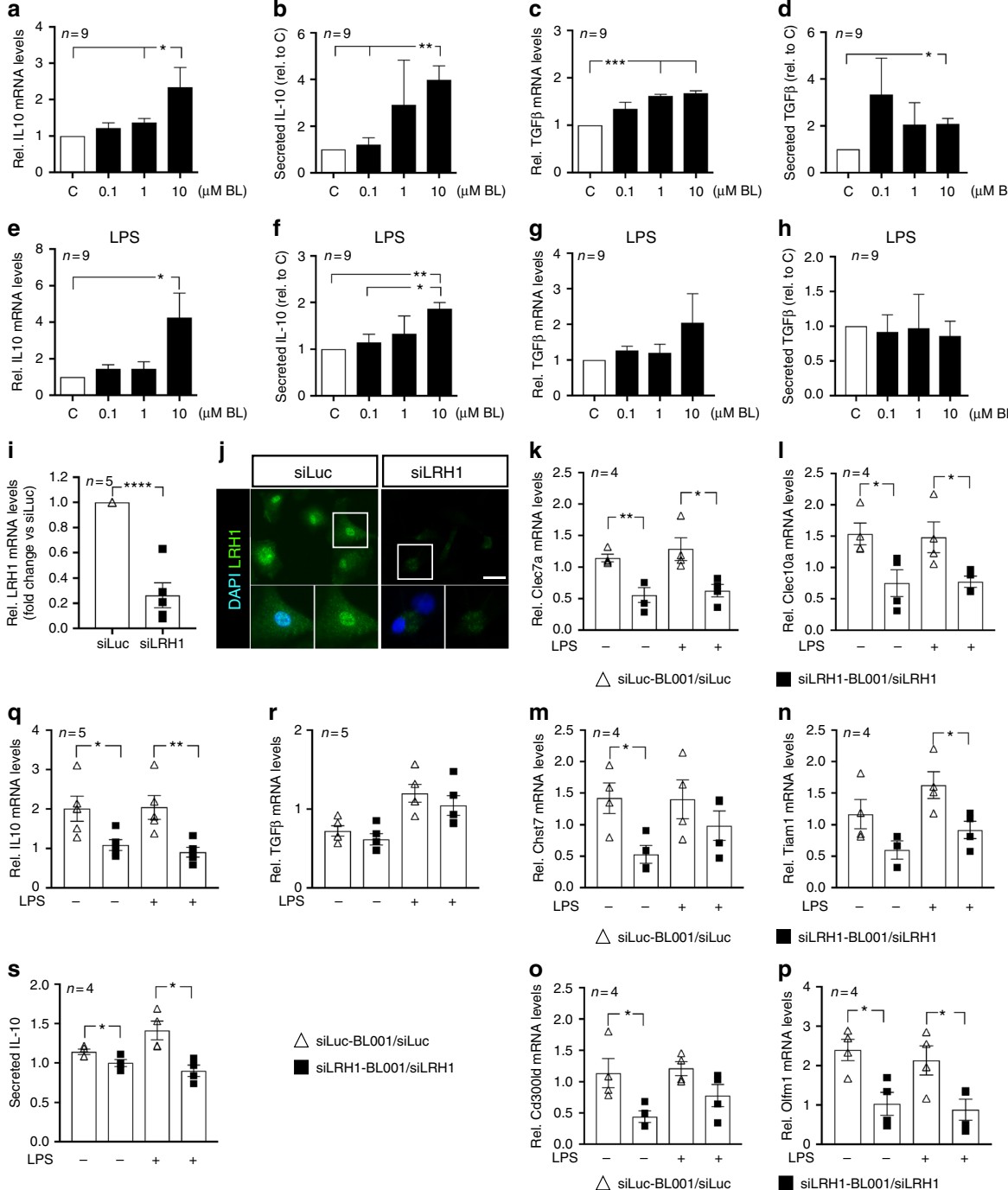

**Fig. 4** LRH-1 is required for BL001-mediated M2 polarization. Primary macrophages were exposed in culture to increasing concentration of BL001 for 24 h. Media was collected and RNA isolated to perform QT-PCR. Transcript and secreted levels of **a**, **b** Il10 and **c**, **d** Tgfb were then determined (n = 9 mice). **e–h** Primary macrophages were incubated for 24 h with increasing concentrations of BL001 alone or in combination with 1.0 µg/ml lipopolysaccharide (LPS). Transcript and secreted levels of **e**, **f** IL10 and **g**, **h** TGFbeta were then measured (n = 9 mice). **i** siRNA-mediated repression of LRH-1 transcript and **j** protein levels in mouse primary macrophages as assessed by QT-PCR and immunostaining (LRH-1, green; scale bar: 25 µM). Relative expression levels of M2 signature genes, **k** Clec7a, **l** Clec10a, **m** Chst7, **n** Tiam1, **o** Cd300ld, and **p** Olfm1 in control or LRH-1-silenced macrophages treated or not with 10 µM BL001 and LPS (n = 4). **q** Il10 and **r** Tgfb transcript levels in siLRH1-treated or not macrophages in the presence or absence of 10 µM BL001 and LPS (n = 5). **s** Secreted levels of IL10 in silenced or not macrophages. Results are expressed as the averages + s.e.m. (**a–i** and **k–s**). *p < 0.05, **p < 0.01, ***p < 0.001, and ****p < 0.0001, one-way ANOVA (**a–h**) and Student's t test (**i**, **k–s**)

mice showed greater numbers of beta cells than the controls (Fig. 6k, l).

**Immune-related biological processes are targeted by BL001**. To evaluate the effects of BL001 on a larger scale, we profiled gene expression via DNA microarray analysis in mouse islets exposed

to BL001 for 2 days. We found that 277, 23, and 195 gene ontology (GO) terms were enriched in islets exposed to 0.1, 1.0, and 10 µM BL001, respectively (Fig. 7a). Seven of these GO terms were common to all conditions, whereas no common down-regulated GO terms was found in the same samples (Fig. 7a). The seven enriched GO terms were related to genes involved in

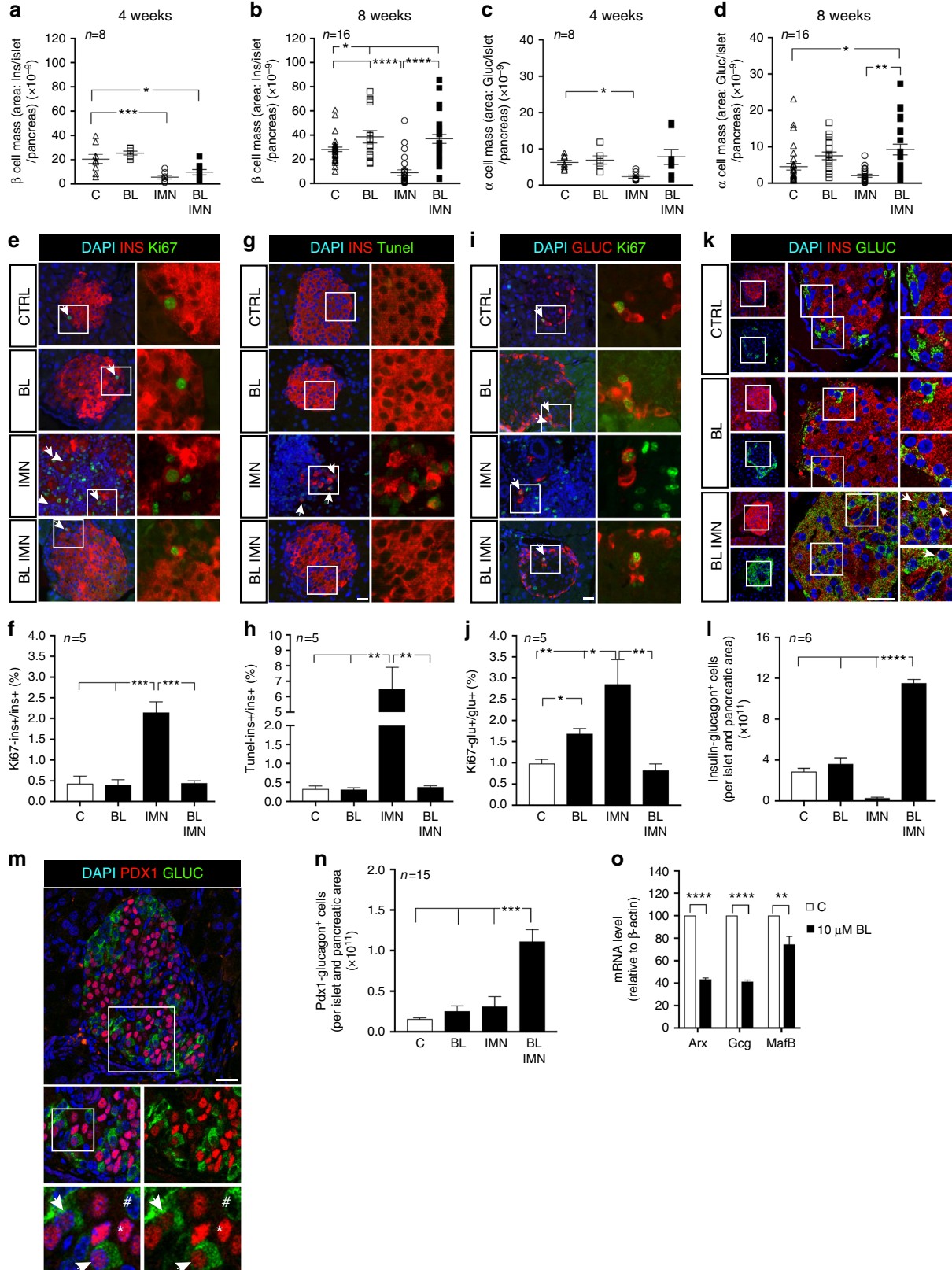

immune functions (Fig. 7b). Heat maps of those genes which were consistently modified, identified 33 genes (Fig. 7c). Those encoding for the CCL2, CCL3, CCL7 chemokines, the cytokine IL6 and the prostaglandin-endoperoxide synthase-2 (PTGS2) were the most induced by the BL001 exposure, whereas the gene coding for the IL1 beta receptor (IL1R1) was the most downregulated (Fig. 7c). Quantitative PCR of these genes, confirmed that the CCL2, CCL3, and PTGS2 transcripts were upregulated by the BL001 exposure (Fig. 7d–f), whereas the IL1R1 transcript was downregulated under the same conditions (Fig. 7g).

## Discussion

In spite of major efforts, the available strategies to restore/preserve a functional beta cell mass through immunomodulation and/or beta cell regeneration/replacement treatments have so far shown limited efficacy for the long-term improvement of glycaemia in T1DM patients[27]. This frustrating situation calls for innovative approaches to this complex problem; identify novel "druggable" targets that could promote the regeneration of a functional beta cell mass, while attenuating the autoimmune attack, and preserving the anti-inflammatory locale that appears necessary for beta cell renewal. Here, we show that BL001, a small agonist of the LRH-1 receptor, has such characteristics, which prevent and revert diabetes in three different mouse models of T1DM.

Our data first document that, by activating LRH-1, BL001 primes macrophages toward the anti-inflammatory M2 phenotype (as revealed by the enhanced M2 genetic signature) resulting in direct stimulation of IL10 expression and secretion[23]. BL001-treated mice also featured increased numbers of pancreatic Tregs, whose expansion is induced by IL10[28], and which are essential in maintaining self-immune tolerance including in T1DM[29]. Circulating levels of CCL2 as well as its expression in islets were increased in BL001-treated and immunized RIP-B7.1 mice. Although the impact of CCL2 as a pro or anti-inflammatory chemokine is disease- and cell-dependent[30], our results suggest that in the context of autoimmune diabetes, increased levels of local CCL2 appears to be associated with the recruitment of macrophages to the pancreas which, in turn are further polarized towards the M2-like subtype under the influence of IL10 production. CCL2 through enhanced IL4 production also fosters an environment that favors expansion of Th2[31], a cell type and cytokine that were increased in the pancreas and in the circulation of immunized, BL001-treated RIP-B7.1 mice. Immunized mice treated with BL001 also displayed increased circulating TGFbeta levels correlating with the presence of a larger number of CD103$^+$/IDO$^+$ dentritic cells, which convey immunosuppressive and tolerogic functions[32,33]. Given that the adoptive

transfer of M2 macrophages as well as the upregulation of CCL2 enhance beta cell survival in NOD mice[34,35] and that the autologous transfer of Tregs improves islet survival and function in T1DM patients[36], the immune changes we observed converge to demonstrate that BL001 significantly inhibits the aggressive autoimmune process favoring tolerance, which presumably contributes to its beneficial effects. In this context, BL001-treated and immunized mice that develop diabetes retained islets with a significant number of PDX1$^+$ cells and reduced insulitis at 8 weeks as compared to immunized mice. Although speculative, the latter results indicate that BL001 dosage or activity was suboptimal in favoring an anti-inflammatory and tolerogenic environment resulting in beta cell destruction and hyperglycemia in these animals.

Blood analysis further showed that BL001 increased circulating levels of the IL5 and IL6 cytokines, and of the chemokine CCL4, which have anti-inflammatory actions. IL6 enhances insulin secretion via the release of GLP1 by alpha cells[37,38] while IL5 and CCL4 shift pro-inflammatory Th1 cells towards the anti-inflammatory Th2 subset and stimulate expansion of Tregs[39,40]. Accordingly, we found reduced numbers of Th1 cells but increased numbers of Th2 and Tregs cells in the pancreas of the immunized and BL001-treated mice. In contrast, circulating levels of IFNgamma, TNFalpha, IL1beta and IL2 remained constant suggesting that production of these cytokines were not directly impacted by the compound. BL001 treatment also stimulated the islet expression of PTGS2, an inducible prostaglandin synthase, whose PGE$_2$ metabolite also inhibits Th1 cells and protects against T1DM[41–43]. In addition, IL6 and the potential secretion of PTGS2-derived PGE$_2$ by islets along with IL4 and IL10 released by Th2 as well as Tregs, will further contribute to M2 polarization within the pancreas independent of the direct activation of LRH-1 in macrophages by BL001[44–47]. These data highlight that, in addition to a direct sizable action on the immune system, BL001 also promotes the release of pancreatic islet-derived factors favoring an anti-inflammatory environment that will further induce tolerogenecity. Future work will focus on dissecting the mechanism whereby the BL001/LRH-1 signaling pathway achieves these beneficial effects.

Our study further documents that the LRH-1 receptor is essential for the proper organization of pancreatic islets. Thus, the beta cell loss of the LHR-1 receptor modifies the proportion of islet cells characterized by an increase in the number of alpha cells, and the retention of a sizable beta cell mass, which associate with a rapid, post-natal death of transgenic mice. Our parallel experiments also show that BL001-mediated activation of LRH-1 stimulates the regeneration of beta cells, in the islets of both control and diabetic mice, an effect which is not attributable to a change in the proliferation and apoptosis of these

**Fig. 5** BL001 promotes beta and alpha cell mass expansion in immunized RIP-B7.1 mice. The control (C) mass of beta (**a**, **b**) and alpha cells (**c**, **d**) of RIP-B7.1 mice was decreased after immunization (IMN), an alteration that was prevented by 10 mg/kg b.w. BL001 daily treatment for 8 weeks (BL IMN). Dot plots show the results from 8 (4 weeks) and 16 (8 weeks) pancreas per group, each dot corresponding to the cell mass of an entire pancreatic section. **e–j** Representative immunofluorescence images of islets co-stained and quantified for **e**, **f** insulin (INS, red) and Ki67 (green), **g**, **h** insulin (red) and Tunel (green), **i**, **j** glucagon (GLUC, red) and Ki67 (green). All values are means of 5 mice per group. Scale bars: 25 µM. Right panels are enlargements of boxed area, and arrow heads point to example of cells simultaneously stained for 2 proteins. **k** Representative confocal images showing the distribution of the insulin- (red) and glucagon-bihormonal cells (green) as well as **l** quantification in islets of the different animal groups. Nuclei were stained with DAPI (blue). Scale bar: 25 µM. Right panel show enlargements of boxed areas. Arrow heads point to cells co-stained for the two depicted markers. **m** A representative image depicting co-immunostaining of PDX1 (red) and glucagon (green) in a pancreas of a normoglycemic immunized BL001-treated mouse, and **n** quantification of such cells in the various groups. Bottom left panels are enlargements of boxed area. Bottom right panels, are without DAPI staining. White arrows point to PDX1$^+$/GLUC$^+$ cells, # identifies a PDX1$^-$/GLUC$^+$ cell and * highlights a PDX1$^+$/GLUC$^-$ cell. $n = 15$ pancreatic slices obtained from five independent mice per group. Scale bar: 25 µM. **o** After a 24-h exposure to 10 µM BL001 (BL), alpha TC-1.6 cells featured decreased expression of the Arx, glucagon (Ggc) and MafB transcripts. Six independent experiments were performed and subsequent QT-PCR conducted in triplicates. Values are means + s.e.m. (**f**, **h**, **j**, **l**, **n**, **o**). *$p < 0.05$, **$p < 0.01$, ***$p < 0.001$, and ****$p < 0.0001$, one-way ANOVA (**a–d**, **f**, **h**, **j**, **l**, **n**) and the Student's $t$ test as compared to control (**o**).

cells. The underlying mechanism may or may not directly target beta cells, and remains to be ascertained. A possible explanation may be an alpha-to-beta cell trans-differentiation[48], as suggested by (1) the frequent occurrence of cells containing both insulin and glucagon after the BL001 administration, (2) repression of the alpha genetic program by BL001, and (3) increased alpha cells subsequent to LRH-1 beta cell-specific deletion. However, this possibility still needs to be validated by specific lineage tracing studies. The fact that BL001 and STZ-treated mice displayed fewer bi-hormonal cells suggest that such trans-differentiation may be facilitated by the increase in

Tregs and M2 macrophages and/or by IL6 and IL10[49,50]. Antecedents for such immune-regenerative mechanism have been described in salamander and zebrafish[51,52]. Furthermore, Tregs and M2 macrophages are key remodeling players in mouse tissue repair (muscle, bone and vasculature) promoting cell differentiation and expansion[53,54].

In summary, our data define LRH-1 as a novel target for the treatment of diabetes, which can be modulated by BL001. The inhibitory effect of the drug on the immune system, its tolerogenic action and its effects on beta cells survival and regeneration combine to account for the beneficial effects observed in rodent

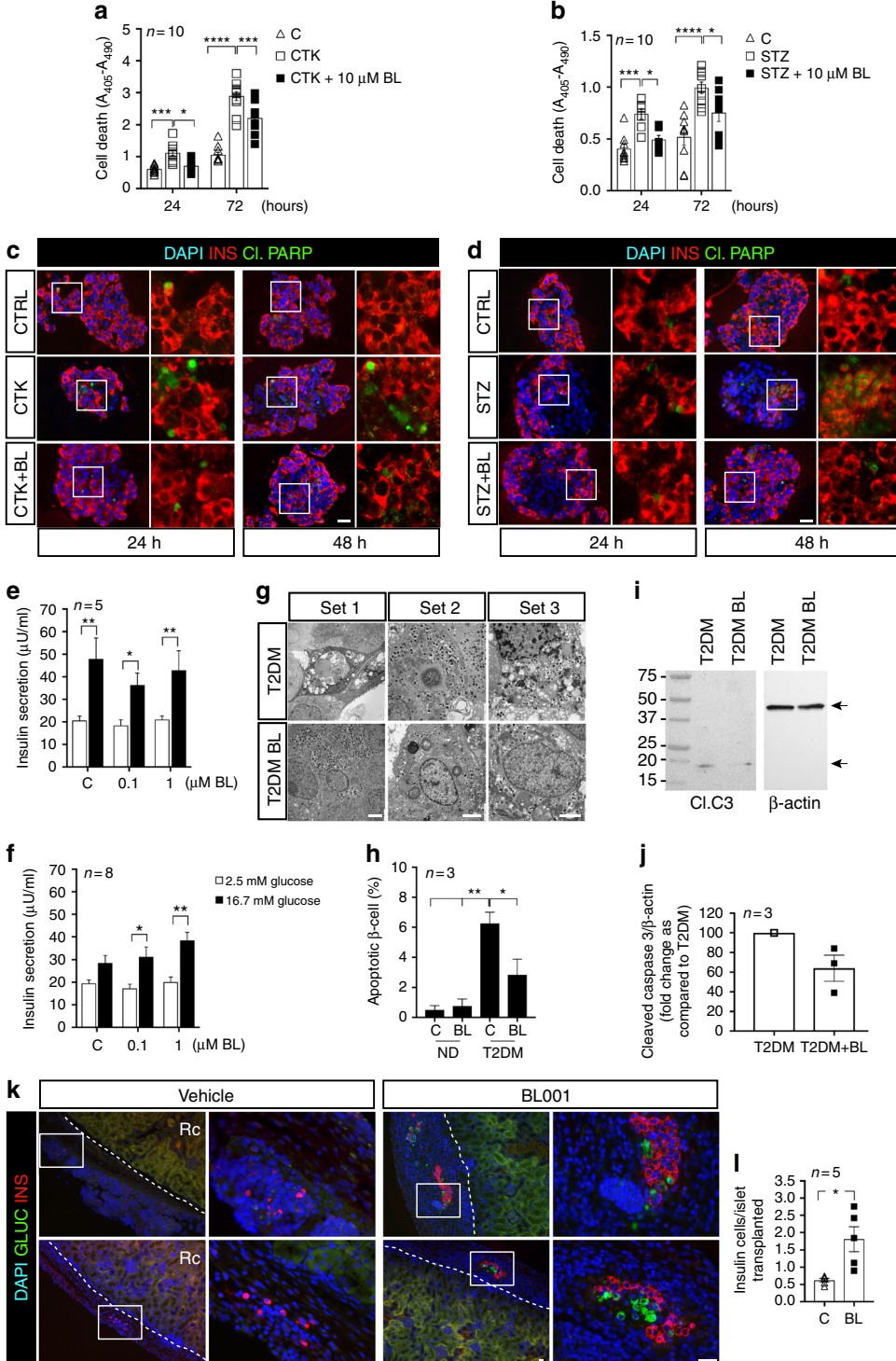

models (Fig. 8). The finding that the drug also decreases beta cell apoptosis in islets from patients with T2DM, and improves their insulin secretion as well as improves beta cell survival in xeno-transplantation, opens the exciting perspective that it could be of value also in the human clinic. In this perspective, the development of new LRH-1 agonists, more stable than BL001 and adapted to oral administration is an urgent need.

## Methods

**BL001 synthesis and formulation**. BL001 [(3aS,6aR)-1,2,3,3a,6,6a-hexahydro-4-(3-methoxyphenyl)-5-((E)-oct4-en-4-yl)-N-phenylpentalen-3a-amine] was synthesized by Sreeni Labs Private Limited (India), at a HPLC purity >98%. The semi-solid compound was dissolved in 100% DMSO, at 0.5 mg/ml and 100 μg/ml stock concentrations for in vitro and in vivo studies, respectively. The optimal formulation for in vivo administration (hereafter referred to as vehicle) was: 1% DMSO, 40% WellSolve (Celeste Corporation, Japan) and 59% water. In vitro pharmacology activity assay and ADME-Tox studies were outsourced to Cerep/Eurofins (http://www.cerep.fr); pharmacokinetic studies were performed by GVK Biosciences Pvt. Ltd. (http://www.gvkbio.com/).

**Mice**. RIP-B7.1[19], LRH-1 Lox/Lox (kindly provided by Dr. K. Schoonjans, EPFL, Switzerland), RIP-Cre[55], C57BL/6 mice (purchased from Janvier Labs, France), and NOD mice (Charles River, Calco Italy) were housed in ventilated plastic cages under a 12-h light/dark cycle, and were given food and water ad libitum. Mice experimentations were approved by the CABIMER Ethics Committee for Animal Experimentation, and performed in accordance with the Spanish law on animal use RD 53/2013. NOD mice studies were carried out in strict accordance with the recommendations in the Guide for the Care and Use of Laboratory Animals of the German Federal Animal Protection Law. Protocols were approved by the Committee on the Ethics of Animal Experiments of the University of Ulm (Tier-forschungszentrum Ulm, Oberberghof) and the Regierungspräsidium Tübingen (Permit Numbers: 1199 and 1327 to R.S.). Mice were randomly distributed for all experiments and not subject to blind analysis. Seven- to eight-week-old males and females RIP-B7.1 mice were injected daily i.p. with 10 mg/kg b.w. BL001, starting either 5 days prior to or after immunization. Immunization was achieved by injection of 50 μg preproinsulin (ppINS) expression plasmid (PlasmidFactory GmbH, Germany) into the two anterior tibialis muscles. Mice were euthanized 4 and 8 weeks after immunization, and pancreases and spleens were extracted for immunocytochemistry and proliferation assays, respectively. In addition, 8-week-old male C57BL/6 mice were treated with BL001 for 5 days prior or after an i.p. injection of 150 mg/kg b.w. streptozotocin (STZ) prepared in 0.01 M sodium citrate at pH 4.5. Twelve-week-old female NOD mice were injected daily i.p. with 10 mg/kg b.w. BL001, for up to 25 weeks. Circulating glucose levels were measured from tail vein blood samples using an Optium Xceed glucometer (Abbott Scientifica SA, Barcelona, Spain). Non-fasting blood glucose ≥13.8 mmol/l for two consecutive measurements was considered to indicate overt diabetes. For insulin tolerance tests (ITT), animals were fasted for 5 h, and then i.p. injected with 0.5 U/kg b.w. Actrapid Insulin. Glycemia was measured at 0, 30, 60, 90, and 120 min after this injection. Cytokine levels were assessed in blood and culture media using the mouse V-PLEX ProInflammatory Panel 1 kit 10-Plex (Meso Scale Discovery, Rockville, USA). Detection was performed by electrochemiluminescence technology, and data acquired on a MSD MESO™ QuickPlex SQ120. Blood triglyceride and cholesterol levels were measured using an Accutrend Plus apparatus (Roche Diagnostics, Mannheim, Germany), using the appropriate strips. Temporal MRI scan of mice were acquired using a Bruker BioSpec 9.4 T/20 animal MRI system, equipped with 400 mT/m gradients and a 40 mm quadrature resonator. Images were acquired using a Turbo-RARTE sequence with respiratory gating (TEeff = 24

ms, TR = 1400 ms, Rare Factor = 4, slice thickness = 0.75 mm, in-plane resolution = 78 × 78 μm).

**Mouse islet isolation**. Mouse islets were isolated by collagenase dissociation, handpicked, and maintained in 11.1 mM glucose/RPMI-1640 (ThermoFisher Scientific, Madrid, Spain) supplemented with 10% fetal bovine serum (FBS; Sigma-Aldrich, Madrid, Spain), 100 U/ml penicillin (Sigma-Aldrich) and 100 mg/ml streptomycin (Sigma-Aldrich)[56].

**Human islet isolation, procuration and treatment**. Human islets were either obtained from The Cell Isolation and Transplantation Center (Department of Surgery, Geneva; Switzerland) or purchased from Tebu-Bio (Barcelona, Spain). Islets from non-diabetic or T2DM organ donors were obtained in Pisa or purchased from Tebu-Bio (Barcelona, Spain). Signed informed consents were obtained from the families of organ donors. The ethical and investigation committee of the University Hospital of Virgen Macarena and Virgen del Rocio approved all procedures (#2013-04398 to B.R.G.). Human islet preparations were washed, hand-picked, and subsequently maintained in CMRL-1066 (ThermoFisher Scientific) containing 5.6 mM glucose, and supplemented with 10% FCS, 100 U/ml penicillin, 100 μg/ml streptomycin, and 100 μg/ml gentamycin (all purchased from Sigma-Aldrich). Human and mouse islets were either untreated or exposed to various concentrations of BL001 for 24–48 h prior to (1) addition of 2 ng/ml IL1beta, 28 ng/ml TNFalpha and 833 ng/ml IFNgamma; (2) addition of 1 mM streptozotocin; (3) assessment of glucose-induced insulin secretion; (4) measurement of apoptosis. In some experiments, LRH1 was repressed in mouse islets and primary peritoneal macrophages by RNA interference. To this end, On-target plus NR5A2 siRNA-smart pool (Dharmacon, cat number L-047044-01) along with either control on-target plus non-targeting pool (Dharmacon, cat number D00181010) for islets or siLuciferase (5′-CGUACGCGGAAUACUUCGA-3′) for macrophages were used in these studies. Fifty nM of siRNAs were pre-mixed with Lipofectamine (Thermo-Fisher Scientific) and subsequently added to cells for 24 h. Fresh medium was then added and cells cultured under various treatments. Apoptosis was measured at 24 and 72 h using the Roche Cell Death Detection ELISA kit (Roche Diagnostics, Mannheim, Germany). This assay is based in the quantitative sandwich-enzyme immunoassay principle using monoclonal antibodies against DNA and histones, that allow specific detection of mono- and oligonucleosomes in apoptotic cells. Islet viability was assessed using the MTT assay, according to the manufacturer's recommendations (Roche, Spain). In some instance, protein extracts were prepared from T2DM islets and cleaved-caspase-3 activity was assessed by western blot analysis[57].

**Human islet transplantation**. Human islet transplantations were performed using a modified protocol from Robertson and Szot[58,59]. Briefly, 10-week-old immuno-competent C57BL/6 male mice were anesthetized by an i.p. injection of 100 mg/kg ketamine and 10 mg/kg xylazine, 150–200 islets were collected without cen-trifugation in a minimum of medium, and transplanted under the kidney capsule using flame-polished borosilicate glass capillaries (Harvard Apparatus, GC100T-10). Upon termination of the experiment, animals were sacrificed and transplanted kidneys extracted, fixed and embedded for further histology analysis. In order to accurately assess transplant reject/engraftment the entire kidney was sectioned and insulin/glucagon co-immunostaining for was performed at every 15th slice, an interval of ~75–150 μm that corresponds to the median size of the majority of islets.

**Cell culture and assays**. The mycoplasma-free alpha TC1–6 cell line was purchased from ATCC (CRL-2934; Barcelona), and maintained in DMEM (ThermoFisher Scientific) supplemented with 10% FBS, 15 mM HEPES, 0.1 mM non-essential amino acids, 0.02% BSA, 2 g/l glucose, 1.5 g/l sodium bicarbonate and 5

**Fig. 6** BL001 protects human islets against apoptosis and rescues insulin secretion in islets of type 2 diabetic donors. **a** A 24 and 72 h exposure to cytokines (CTK) or **b** streptozotocin (STZ) increased cell death in human islets. This effect was prevented by 10 μM BL001 (BL). n = 10 independent islet preparations, performed in either duplicates or triplicates. Double immunostaining for insulin (INS) and cleaved PARP (Cl.PARP) in sections from human isolated islets exposed to either **c** cytokine (CTK) or **d** streptozotocin (STZ) and treated or not with 10 μM BL001 (BL). Scale bars: 25 μM. Right panels of each time point are enlargements of boxed area in left panel. **e** After a 24-h exposure to 0 (C) −1 μM BL001, islets from normoglycemic donors (n = 5 independent donors) were similarly stimulated by glucose to release insulin. **f** In contrast 0.1–1 μM BL001 increased glucose-stimulated insulin secretion in islets of type 2 diabetic donors (n = 8 independent donors). Data are means + s.e.m. **g** Electron micrographs show apoptotic beta cells in islets of type 2 diabetic donors (T2DM), which were unfrequent when these islets were exposed to 1 μM BL001 (T2DM BL). Scale bars: 2 μM. **h** Quantification shows that the percentage of apoptotic beta cells decreased after exposure to 1 μM BL001, in three independent donors. **i** Western blot of cleaved caspase-3 (Cl.C3) and actin in protein extracts isolated from control (T2DM) and BL001 treated T2DM islets (T2DM BL) along with **j** densitometric analysis. n = 3 independent donors. **k** 10-week-old C57BL/6 male mice were transplanted with 150–200 human islets under the kidney capsule. Two days after transplantation, mice were treated daily with BL001 or vehicle for seven days. Kidneys were harvested and processed for immunofluorescence. Two independent representative image sets of kidney sections immunostained with sera against insulin (red) and glucagon (green). DAPI nuclear counterstaining is used. Scale bar: 25 μM. Right panels correspond to the white squares indicated in the left panels. Rc Renal cortex. **l** Quantification of insulin-positive cells in islets transplanted under the kidney capsule. n = 5 independent transplantations. *p < 0.05, **p < 0.01, ***p < 0.001, and ****p < 0.0001, two-way ANOVA (**a**, **b**) and Student's t test (**e**, **f**, **h**, **j**, **l**)

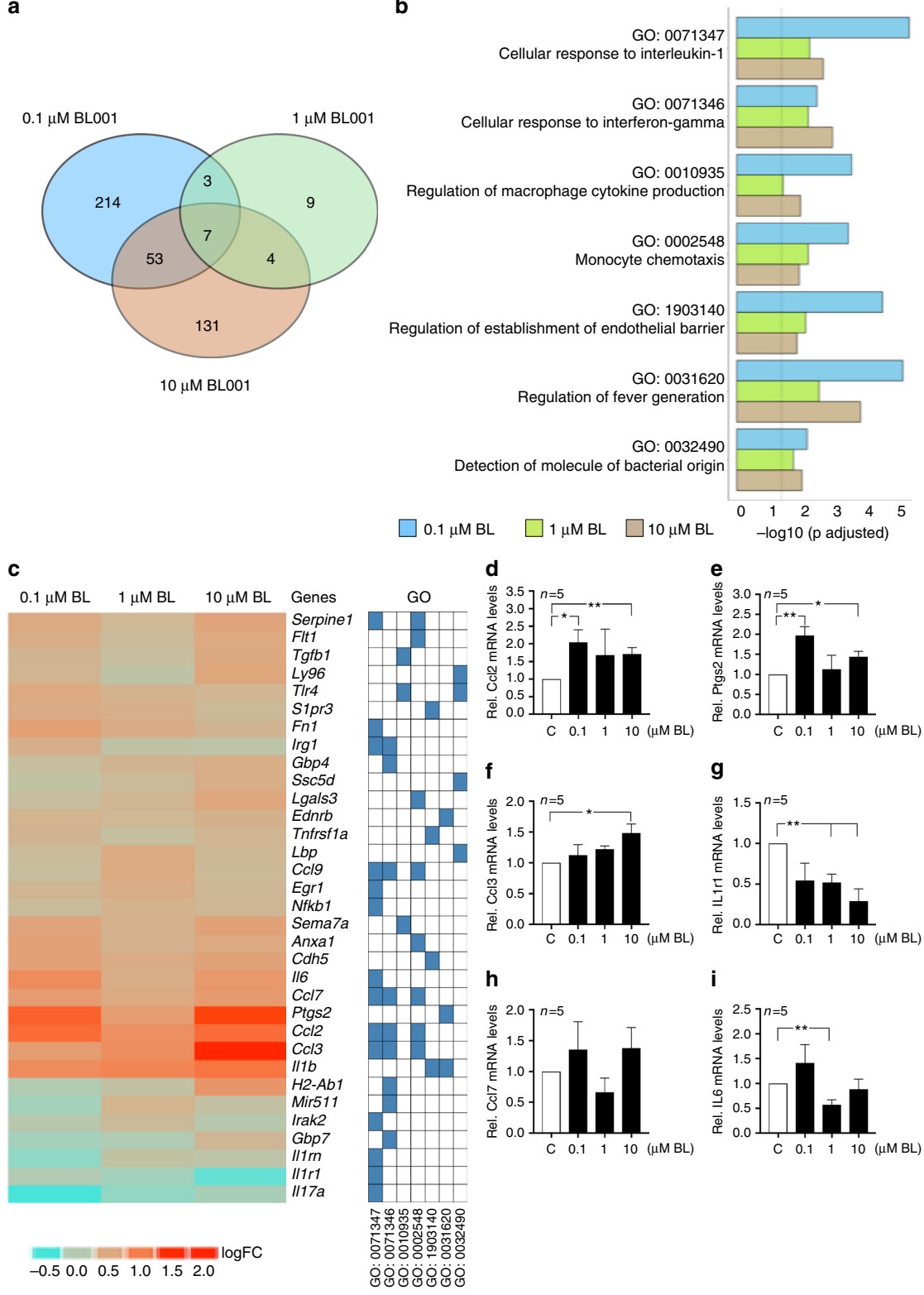

**Fig. 7** BL001 potentiates islet genes involved in immunomodulation. Isolated mouse islets were treated with 0.1–10 μM BL001 for 24 h and RNA isolated for DNA microarray analysis ($n = 3$ independent islet preparations per BL001 concentrations). **a** Venn diagram depicting the number of GO terms significantly enriched after three different BL001 treatment. **b** Enrichment plot for the seven GO terms common to all three BL001 concentrations. **c** Heatmaps displaying logFC values of transcripts modulated by BL001, and their association with common GO processes. Blue cells reflect the membership of a gene to a given GO BP term. Validation by QT-PCR of **d** Ccl2, **e** Ptgs2, **f** Ccl3, **g** Il1r1, **h** Ccl7, and **i** Il6 transcript levels in mouse islets treated with BL001. $n = 5$ independent islet preparations. *$p < 0.05$ and **$p < 0.01$, Student's $t$ test versus control untreated (C)

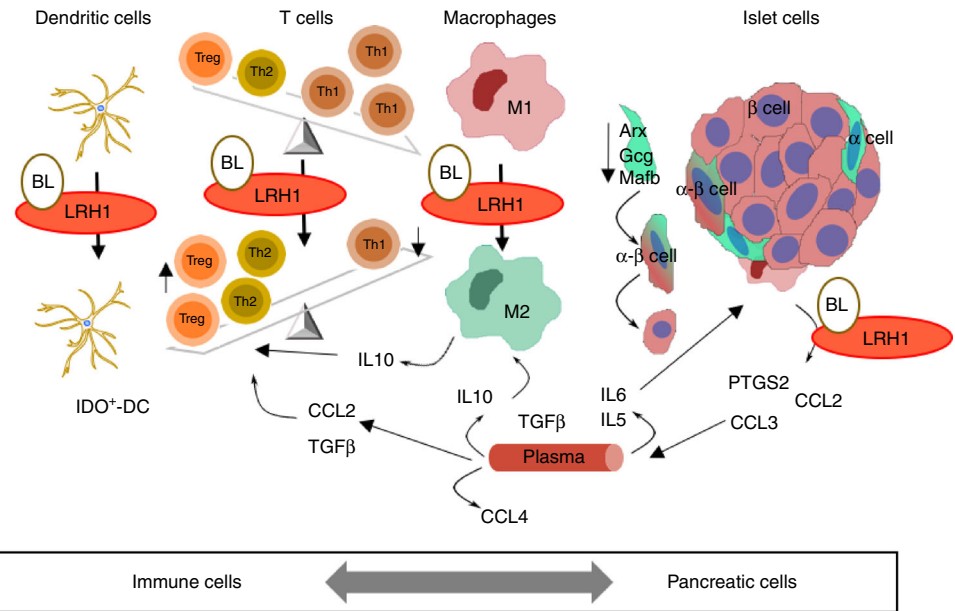

**Fig. 8** Proposed model of BL001/LRH-1 cellular action. Schematic view of the effects of BL001 in changing the pancreas pro-inflammatory immune environment toward an anti-inflammatory environment, which promotes beta cell regeneration, possibly through alpha-to-beta cell trans-differentiation

μM beta mercaptoethanol (all purchased from Sigma-Aldrich). The proliferation of splenocytes isolated from mouse spleens was assessed after a 3-day culture in RPMI 1640 medium supplemented with 8% FBS, 20 mM L-glutamine, 1% sodium pyruvate, 1% nonessential amino acids, and 1% penicillin/streptomycin (all from Invitrogen), in the presence or absence of the insulin peptide SLYQLENYCA. Cells were pulsed with [³H]-thymidine for the last 24 h of culture, harvested and lysed onto membranes prior to liquid scintillation counting using a Beckman Coulter LS 6500 counter. Mouse primary macrophages were isolated from the peritoneal cavity, and cultured in DMEM/F12–10 (ThermoFisher Scientific) supplemented with 10% FBS, 2 mM L-glutamine, 100 U/ml penicillin, 100 μg/ml streptomycin (all purchased from Sigma-Aldrich). Cells were stimulated with 1 μg/ml LPS (in DMSO), in the absence or presence of 0.1, 1, or 10 μM BL001 for 24 h. The secretion of cytokines was measured in the culture medium by electro-chemiluminescence technology from MesoScale Discovery (Rockville, USA), and RNA was extracted from cells.

**Flow cytometry.** Subpopulations of T helper cells extracted from mouse pancreas, were characterized by flow cytometry (FACSCalibur, BD Biosciences, Madrid, Spain), using the following antibodies (Supplementary Table 3): FITC-conjugated anti-mouse CD4; Alexa fluor 647-conjugated anti-mouse IL17; Alexa fluor 647-conjugated anti-mouse IL4; PE-Cy7-conjugated anti-mouse IFNgamma. Regulatory T cells were evaluated using FITC-conjugated anti-mouse CD4, in combination with APC-conjugated anti-mouse CD25, and PE-conjugated anti-mouse FoxP3 antibodies. Pancreatic macrophage subpopulations were assessed using Alexa fluor 488 anti-mouse CD45, BV421 anti-mouse CD11b, APC anti-mouse F4/80 and PE anti-mouse CD206. Dendritic cells were evaluated using Alexa fluor 488 anti-mouse CD45, PE anti-mouse CD103 and Alexa fluor 647 anti-IDO. Data were analyzed using FlowJo V9 software (Tree Star). Cell sorting was performed using a FACSAria I (BD).

**Immunohistochemistry and electron microscopy.** Pancreases and isolated islets were fixed and embedded as previously detailed[56]. Primary and secondary antibodies are listed in Supplementary Table 3. Nuclear counterstaining was performed by DAPI, and sections were mounted using DAKO fluorescent mounting medium. Islet cell apoptosis was measured using a TUNEL assay (In Situ Cell Death Detection Kit, Roche, Madrid, Spain). Images were acquired using either a Leica DM6000B or a Leica TCS SP5 confocal microscope. For the assessment of beta- and alpha-cell mass, images of pancreatic sections were automatically acquired using a software (NIS-Elements imaging)-controlled data acquisition Nikon eclipse Ti-e microscope (Nikon). Morphometric quantification was performed using the Fiji/ImageJ software. Insulitis was scored in paraffin sections of pancreas, stained with H&E. Cells with small nuclei were considered of haematopoietic origin. Insulitis scoring was performed as a grade of 0 to 4 according to percentage of infiltrated islet area (0, 0%; 1<10%; 10%<2>55%; 55%<3>75%; 4>75%). For electron microscopy, pancreatic islets were processed using a standard Spurr protocol[60]. Electron microscopy images were acquired with an EMCCD camera (TRS 2kx2k). The number of non-apoptotic and apoptotic (visualized by blebbing and nuclear condensation) beta cells was counted and the respective percentage of

dying cells was expressed as the number of apoptotic cell type over the total number of beta cells.

**DNA microarray.** Labeled cRNA samples were prepared from pools of at least 100 islets isolated from 8-week-old C57BL/6 female mice, treated or not with increasing concentrations of BL001, and subpopulations of M2-like macrophages (CD45 +/F4/80+/CD11b+/CD206+) purified from the pancreas of either vehicle- or BL001-treated and immunized C57BL/6 mice. Three independent preparations of cRNA were prepared per group, and hybridized to the GeneChip Mouse Gene 2.0 ST Array (islets) and to the Clariom S Assay Mouse Array (M2-like macrophages) (Affymetrix, Santa Clara, CA), using standard protocols of the Genomic Core Facility of CABIMER. For each microarray experiment, the Robust Multiarray Analysis (RMA) method was applied on a per-chip basis for background correction[61]. Subsequent normalization across arrays, and summarization were performed using a quantile algorithm and median-polish, respectively[62], via oligo package from Bioconductor (http://www.bioconductor.org). A differential gene expression analysis was then performed using the limma package[63]. Computed $p$ values were corrected using the false discovery rate (FDR) method, to harmonize for the multiple comparisons of all genes[64].

For the islet samples, gene set analysis was performed using the logistic regression model[65], while GO annotation for genes in the microarrays were extracted from Bioconductor GO.db annotation package[66]. Heatmaps of logFC values from differential expression analyses (different concentrations of BL001 versus control) were generated for those genes which were expressed in all groups (raw $p$-value < 0.05), and associated to the statistically enriched ($p$-adjusted <0.05) seven GO terms identified at all drug concentrations.

For the M2-like macrophages, a heatmap of log2 expression values of a subset of M2 gene signatures[21], was generated for the CD45+/F4/80+/CD11b+/CD206+ subpopulation isolated from either untreated (IMN) or BL001-treated immunized (IMN BL) mice. LogFC values from differential expression analyses (IMN BL versus IMN) were generated for these genes (raw $p$-value < 0.05).

**RNA isolation and quantitative PCR (QT-PCR).** Total RNA was extracted using the RNeasy Micro Kit (Qiagen). Complementary DNA using 0.5 to 1 μg RNA was synthesized using the Superscript II (ThermoFisher Scientific). The RT-PCR was performed on individual cDNAs using SYBR green (Roche)[56]. Primers can be obtained upon request.

**Statistical analysis.** The Ruth Lenth's power of analysis was applied to the different animal models to ensure that adequate numbers of animals had been studied to detect significant changes. Results are expressed as mean ± s.e.m. (line plots as a function of time) or as mean + s.e.m. (bar graphs). Statistical analysis were performed using the GraphPad Prism software (GraphPad Software, La Jolla, USA). Statistical differences were estimated by one- or two-way ANOVA, with Bonferroni post hoc tests, Student's $t$ test or non-parametric Mann–Whitney test, whichever was appropriate.

**Data availability**. The Microarray raw data that support the findings of this study have been deposited in the Gene Expression Omnibus repository with the super-series accession number GSE94505 and subseries numbers GSE94505 and GSE104322.

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

## Acknowledgements

We thank Maria Torres, Evie Egelmeers, and Daniel Parras Molina for excellent technical assistance as well as Drs. Maria José Quintero, Paloma Dominguez, Marta Cejudo Guillén, and Soledad López Enriquez for analysis support. This work was funded by grants from the Juvenile Diabetes Research Foundation (17-2013-372 to B.R.G.), the Consejeria de Salud, Fundacion Publica Andaluza Progreso y Salud, Junta de Andalucia (PI-0727-2010 to B.R.G. and P10CTS6505 to B.S.), Consejeria de Economia, Innovacion y Ciencia (P10.CTS.6359 to B.R.G.), the Ministerio de Economia y Competidivad co-funded by Fondos FEDER (PI10/00871, PI13/00593, and BFU2017-83588-P to B.R.G.; PI14/01015, RD12/0019/0028, and RD16/0011/0034 to B.S.; PI16/00259 to A.H.) and Deutsche Forschungsgemeinschaft (GRK-1789 ´CEMMA´ and DFG SCHI-505/6-1 to R.S.). Special thanks to the families of the DiabetesCero Foundation that graciously supported this work (to B.R.G.). A.M.M. is a recipient of a Miguel Servet grant (CP14/00105) from the Instituto de Salud Carlos III co-funded by Fondos FEDER whereas E.F.M. is a recipient of a Juan de la Cierva Fellowship. I.G.H.G. is supported by a fellowship from Amarna Therapeutics. In some instances, human islets were procured through the European Consortium for Islet Transplantation funded by Juvenile Diabetes Research Foundation (3-RSC-2016-162-I-X).

## Author contributions

N.C.V. and B.R.G. conceptualized the study. N.C.V., B.R.G., and P.M. wrote the manuscript. N.C.V., P.I.L., P.M., L.S.P., L.S.O., F.J.B.S., and B.R.G. designed the study and analyzed the data. N.G.R., I.G.H.G., E.F.M., L.L.N., S.Y.R.Z., M.B., C.C.L., K.S., M.B., D.B., G.P., D.P., M.G.T, V.dT., R.S., G.P., L.S.P., P.dH., A.H., J.M.M.G., B.S., and A.M.M. were involved in the acquisition and analysis of data. K.Sc. supplied transgenic mice. Bioinformatics analysis was performed by J.P.F. All authors critically reviewed the manuscript for important intellectual content and approved the final version to be published. B.R.G. is the guarantor of this work.

## Additional information

**Competing interests:** Two patents (WO 2011 144725 A2 and WO 2016 156531 A1) related to BL001 have been published, of which B.R.G., N.C.V., D.P., L.S.O., and M.B. are inventors. These patents have been licensed to ARIDDAD Therapeutics, a Biotech spinoff co-founded by B.R.G., N.C.V., L.St.-O., and P.dH. The remaining authors declare no competing interests.

