## [Peer review file · Nature Communications]

Reviewers' comments:

Reviewer #1 (Remarks to the Author):

In response to our comments concerning the specificity of LRH-1 activation by BL001 in macrophage polarization, the authors have made efforts to characterize the phenotype of BL001-activated LRH-1-silenced macrophages. Indeed, they compared the mRNA levels of Clec7a, Clec10a, Olfm1, Chst7, Tiam1 and Cd300ld, specific markers of M2 polarization between BL001-activated macrophages versus BL001-activated LRH-1-silenced macrophages. Interestingly, the authors showed that the silencing of LRH-1 in macrophages significantly reduced the BL001-mediated increase of Clec7a, Clec10a, Olfm1, Chst7, Tiam1 and Cd300ld and Il10 expression, further supporting the role of LRH-1 in BL001-induced M2 macrophage differentiation.

As indicated in the discussion section, future works will be essential to dissect the immune mechanism whereby the BL001/LRH-1 signaling pathway achieves an anti-inflammatory environment that will further induce tolerogenicity.

In conclusion, the authors have carried out further investigation to satisfy the referees' concerns about the role of BL001 in macrophage polarization through LRH-1. The current version of this manuscript contains improved data and better discussed.

Reviewer #2 (Remarks to the Author):

Authors provide a convincing case that the LRH1 agonist, BL-001, provides protective effects in mouse beta cells in response to cytokines, STZ, and immune attack in vivo, and which appears to reflect components of beta cell survival as well as immunoprotection, and extends to xenotransplanted human islets in vivo into mice. My comments have been adequately addressed. The items below should be addressed.

Major Comments.

1. In Fig 5g, the PDX1 immunostaining pattern is strange: it is a nuclear protein, but appears diffusely cytoplasmic and speckled here. I do not think these data are reliable. It would be important to show controls in a PDX1-null cell type. Without this, it is hard to claim authentic PDX1-glucagon co-staining. Thus, Figure 5g should be deleted along with Fig 5h, and with the relevant text in the Results, Discussion and Figure 5 Legend. Alternately, the authors must provide more extensive immunocytochemistry data - with unequivocal positive and negative controls - for a series of beta cell markers including insulin, PDX1, MAFA, NKX6.1 and alpha cells, including ARX1, GC and glucagon.

2. On lines 269-270 we are told that "complete rejection" has occurred, but Figure 6h clearly shows residual beta cells, as does the accompanying bar graph. I think it is fair, overall, to say that the grafts from the BL001-treated animals shows greater numbers of beta cells than the controls, although it is impossible to know if this reflects differences in survival, proliferation, and/or transdifferentiation.

Minor Comments.

3. Line 136. Fig 1 a-d. Indicate how cell death was measured in Methods and Fig 1a-d Fig legs. Simply saying "by Roche Kit" is not adequate. Indicate what the assay measures in the Methods and Figure Legend.
4. In Fig 1b-d legends, define the term "siSC" (I assume silencing control, or siRNA against scrambled sequence?) and provide sequences and vectors/viruses types for silencing experiments.
5. Similarly, in Fig 4, the vectors and sequences used to silence LRH-1 and the si-luciferase control should be described and sequences provided.
5. On lines 252-255 referring to Suppl Fig 10a/10b, I cannot find the "metabolic data". I suspect this is left over from an earlier draft.

Point-by-point response to comments of referee 2

We thank the Editors for their interest in our work. To their request we have addressed the specific concerns raised by reviewer 2, taking into account their recommendations regarding concern 1.

Reviewers' comments:

Reviewer #1 (Remarks to the Author):

In conclusion, the authors have carried out further investigation to satisfy the referees' concerns about the role of BL001 in macrophage polarization through LRH-1. The current version of this manuscript contains improved data and better discussed.

Authors reply:

We thank the reviewer for his/her endorsement.

Reviewer #2 (Remarks to the Author):

Authors provide a convincing case that the LRH1 agonist, BL-001, provides protective effects in mouse beta cells in response to cytokines, STZ, and immune attack in vivo, and which appears to reflect components of beta cell survival as well as immunoprotection, and extends to xenotransplanted human islets in vivo into mice. My comments have been adequately addressed. The items below should be addressed.

Authors reply:

We thank the reviewer for highlighting the fact that we have adequately addressed his/her concerns from several previous round of reviews.

Major Comments.

1. In Fig 5g, the PDX1 immunostaining pattern is strange: it is a nuclear protein, but appears diffusely cytoplasmic and speckled here. I do not think these data are reliable. It would be important to show controls in a PDX1-null cell type. Without this, it is hard to claim authentic PDX1-glucagon co-staining. Thus, Figure 5g should be deleted along with Fig 5h, and with the relevant text in the Results, Discussion and Figure 5 Legend. Alternately, the authors must provide more extensive immunocytochemistry data - with unequivocal positive and negative controls - for a series of beta cell markers including insulin, PDX1, MAFA, NKX6.1 and alpha cells, including ARX1, GC and glucagon.

The authors sought the Editors advice on this comment as it intrudes with a prior recommendation from reviewer 3 to specifically quantify these cells.

Editors reply: *We suggest providing a clearer immunofluorescence for fig.5g and more representative images in a supplementary figure.*

Authors reply:

Following the recommendation of the Editors, we now provide improved confocal immunofluorescence images depicting an islet with at least 2 glucagon cells clearly expressing PDX1 within the nuclei (new **Fig 5g**). We also provide 3 additional representative confocal images in a new **Supplementary Fig. 10**. The 'speckled' effect maybe a consequence of lower PDX1 expression levels in the GLUC⁺ cells. In addition, several independent studies have shown that under stress condition PDX1 can undergo nucleo-cytoplasmic shuttling an effect that is mediated by FOXO1^{1,2}. Thus, PDX1 is not exclusively nuclear.

2. On lines 269-270 we are told that "complete rejection" has occurred, but Figure 6h clearly shows residual beta cells, as does the accompanying bar graph. I think it is fair, overall, to say that the grafts from the BL001-treated animals shows greater numbers of beta cells than the controls, although it is impossible to know if this reflects differences in survival, proliferation, and/or transdifferentiation.

Authors reply:

We have revised this sentence to:

At this time point, rejection of human islets is anticipated in control mice^{3,4}. Consistent with the protective effect of BL001, grafts from BL001-treated mice showed greater numbers of beta cells than the controls (Fig. 6h).

Minor Comments.

3. Line 136. Fig 1 a-d. Indicate how cell death was measured in Methods and Fig 1a-d Fig legs. Simply saying "by Roche Kit" is not adequate. Indicate what the assay measures in the Methods and Figure Legend.

Authors reply:

We now include a brief description of cell death measurement using the 'Roche kit' in the Methods section (page 16):

This assay is based in the quantitative sandwich-enzyme immunoassay principle using monoclonal antibodies against DNA and histones, respectively, that allow specific detection of mono- and oligonucleosomes in apoptotic cells.

and

Figure 1 legend:

Cell death was assessed by ELISA quantification of mono and oligonucleosomes released by apoptotic cells.

4. In Fig 1b-d legends, define the term "siSC" (I assume silencing control, or siRNA against scrambled sequence?) and provide sequences and vectors/viruses types for silencing experiments.

Authors reply:

We now clarify in Fig. 1 legend that siSC refers to siRNAs targeted to scrambled sequences. We also provide references for all siRNA sequences in the Methods section (page 16).

5. Similarly, in Fig 4, the vectors and sequences used to silence LRH-1 and the si-luciferase control should be described and sequences provided.

Authors reply:

References for all siRNA sequences are provided in the Methods section (page 16).

6. On lines 252-255 referring to Suppl Fig 10a/10b, I cannot find the “metabolic data”. I suspect this is left over from an earlier draft.

Authors reply:

We have removed ‘metabolic data’ and revised the sentence to:

Ten to 20 μ M BL001 did not reveal any cytotoxic effects on islets obtained from normoglycaemic donors (Supplementary Fig. 11a)

References

1. Kawamori, D., *et al.* The forkhead transcription factor Foxo1 bridges the JNK pathway and the transcription factor PDX-1 through its intracellular translocation. *J Biol Chem* **281**, 1091-1098 (2006).
2. Meng, Z., *et al.* Forkhead box O1/pancreatic and duodenal homeobox 1 intracellular translocation is regulated by c-Jun N-terminal kinase and involved in prostaglandin E2-induced pancreatic beta-cell dysfunction. *Endocrinology* **150**, 5284-5293 (2009).
3. Brandhorst, D., Brandhorst, H., Maataoui, V., Maataoui, A. & Johnson, P.R. Anti-caspase-3 preconditioning increases proinsulin secretion and deteriorates posttransplant function of isolated human islets. *Apoptosis* **18**, 681-688 (2013).
4. Qi, M., *et al.* Survival of human islets in microbeads containing high guluronic acid alginate crosslinked with Ca²⁺ and Ba²⁺. *Xenotransplantation* **19**, 355-364 (2012).

REVIEWERS' COMMENTS:

Reviewer #2 (Remarks to the Author):

My questions have been addressed adequately.